# Zebrafish CCNF and FUS Mediate Stress-Specific Motor Responses

**DOI:** 10.3390/cells13050372

**Published:** 2024-02-21

**Authors:** Yagiz Alp Aksoy, Alexander J Cole, Wei Deng, Daniel Hesselson

**Affiliations:** 1Sydney Medical School, Faculty of Medicine and Health, The University of Sydney, Sydney, NSW 2006, Australia; yagiz.aksoy@health.nsw.gov.au; 2Cancer Diagnosis and Pathology Group, Kolling Institute of Medical Research, Royal North Shore Hospital, St Leonards, NSW 2065, Australia; 3Macquarie Medical School, Macquarie University, Sydney, NSW 2109, Australia; 4Centenary Institute, Faculty of Medicine and Health, The University of Sydney, Sydney, NSW 2006, Australia; a.cole@centenary.org.au; 5School of Biomedical Engineering, University of Technology Sydney, Ultimo, NSW 2007, Australia; wei.deng@uts.edu.au

**Keywords:** Amyotrophic Lateral Sclerosis (ALS), CCNF gene, FUS gene, Zebrafish models, CRISPR/Cas9 genome editing, TALEN-mediated genome editing, motor neuron development, oxidative stress response, endoplasmic reticulum stress, neurodegenerative disease research, genetic mutations in ALS, motor function analysis, Zebrafish embryonic development, stress-induced motor impairment, ALS pathogenesis, CRISPR, TALEN, in vivo genome editing

## Abstract

Amyotrophic lateral sclerosis (ALS) is a devastating neurodegenerative disease characterized by the degeneration of motor neurons. Mutations in the cyclin F (*CCNF*) and fused in sarcoma (*FUS*) genes have been associated with ALS pathology. In this study, we aimed to investigate the functional role of CCNF and FUS in ALS by using genome editing techniques to generate zebrafish models with genetic disruptions in these genes. Sequence comparisons showed significant homology between human and zebrafish CCNF and FUS proteins. We used CRISPR/Cas9 and TALEN-mediated genome editing to generate targeted disruptions in the zebrafish *ccnf* and *fus* genes. Ccnf-deficient zebrafish exhibited abnormal motor neuron development and axonal outgrowth, whereas Fus-deficient zebrafish did not exhibit developmental abnormalities or axonopathies in primary motor neurons. However, Fus-deficient zebrafish displayed motor impairments in response to oxidative and endoplasmic reticulum stress. The Ccnf-deficient zebrafish were only sensitized to endoplasmic reticulum stress, indicating that ALS genes have overlapping as well as unique cellular functions. These zebrafish models provide valuable platforms for studying the functional consequences of CCNF and FUS mutations in ALS pathogenesis. Furthermore, these zebrafish models expand the drug screening toolkit used to evaluate possible ALS treatments.

## 1. Introduction

Amyotrophic lateral sclerosis (ALS) is a devastating neurodegenerative disease characterized by the progressive loss of motor neurons, resulting in muscle weakness, paralysis, and, ultimately, respiratory failure. Although the precise etiology of ALS remains elusive, it is well-established that genetic mutations play a significant role in its pathogenesis. Among the numerous genes implicated in ALS, cyclin F (*CCNF*) and fused in sarcoma (*FUS*) have emerged as key players in the disease.

*CCNF* encodes a regulatory protein involved in cell cycle control and protein degradation pathways. Recent studies have shown that mutations in *CCNF* are associated with ALS, both in familial and sporadic cases [1]. Mutated cyclin F exhibits aberrant protein degradation functions, leading to the accumulation of misfolded proteins and impaired cellular homeostasis [2]. This disruption of protein quality control mechanisms contributes to neuronal dysfunction and degeneration in ALS [3].

Similarly, *FUS*, which encodes the fused in sarcoma protein, is a RNA-binding protein involved in the regulation of multiple cellular processes, including transcriptional regulation, RNA processing, and transport, that have been linked to ALS [4,5]. Mutations in *FUS* have been identified in both familial and sporadic ALS cases, highlighting its significance in disease pathogenesis [6,7]. Pathogenic *FUS* mutations lead to the mislocalization and aggregation of the protein, impairing its normal functions and causing cellular toxicity [8].

Animal models have an instrumental role to play in unravelling the underlying mechanisms and functional consequences of *CCNF* and *FUS* mutations in ALS. Zebrafish (*Danio rerio*) has emerged as a powerful model organism for studying neurodegenerative diseases due to its genetic tractability, optical transparency, and conserved neurodevelopmental processes [9,10]. Utilizing genome editing tools, such as CRISPR/Cas9 (clustered -regularly-interspaced-short palindromic repeats/CRISPR-associated protein 9) and TALEN (transcription-activator-like effector nucleases), allows precise genetic modifications in zebrafish, enabling the generation of targeted disease models [11,12,13].

The use of genome editing tools offers several advantages to the study of the functional roles of CCNF and FUS in disease [14]. Firstly, these techniques enable the generation of zebrafish models with specific mutations in the targeted genes, recapitulating the genetic alterations found in ALS patients. This approach facilitates the investigation of the causal relationship between *CCNF/FUS* mutations and ALS pathology, providing insights into the molecular mechanisms underlying motor neuron degeneration. Furthermore, genome editing tools allow for the assessment of the functional consequences of *CCNF* and *FUS* mutations in vivo. 

Here we generated zebrafish models with disrupted *Ccnf* or *Fus* genes, to examine the effects on motor neuron development, axonal outgrowth, and motor behaviors. The ability to observe these phenotypic changes in a living organism provides a valuable platform to study the impact of *CCNF* and *FUS* mutations on the nervous system in organisms exposed to defined stressors.

## 2. Methods

### 2.1. Zebrafish Husbandry

The husbandry, breeding, and mating of wildtype and mutant zebrafish adhered to established procedures [15]. Embryos and larvae were maintained at a constant temperature of 28.5 °C in E3 medium supplemented with methylene blue, until they reached 5 days post-fertilization (dpf), and were then transferred to dedicated tanks. To facilitate the removal of chorions, a solution of 10 μL pronase was gently introduced to embryos within Petri dishes, commencing at 1 dpf. This chorion removal process was exclusively applied to embryonic stages that were analyzed before reaching 3 dpf. In the earliest stages (prior to 2 dpf), the chorions were delicately detached using forceps. All zebrafish embryos and adult individuals were cared for and handled in accordance with standard protocols approved by the Animal Research Ethics Committee at Macquarie University (ARA: 2015-034).

### 2.2. Target Site Selection & Production of sgRNA

The selection of target sites for sgRNA production was conducted by following the methodologies outlined in Aksoy et al. 2019 [16] and Aksoy et al. 2020 [17]. Primers were listed in Appendix A.

### 2.3. Zebrafish-Specific Optimizations for Cas9-Mediated Mutagenesis

To optimize efficient mutagenesis by NHEJ-mediated indels in zebrafish, we implemented zebrafish-specific optimizations [16]. We first compared co-injection of sgRNA and Cas9 protein, instead of Cas9 mRNA. We found direct injection of active Cas9 protein was 1.8-fold more effective in generating indels at the target site, compared to Cas9 mRNA (Appendix A; Cas9 mRNA, 29.56%, *n* = 100; Cas9 protein, 54.21%, *n* = 100). Furthermore, we did not find any difference between the survival of embryos injected with Cas9 protein or Cas9 mRNA (Appendix A). In this experiment, we initially identified 14 DNA targets on exon-16 of zebrafish *ccnf* gene. We excluded 6 targets due to the predicted possible off-targets in zebrafish genome. We determined the cutting efficiency of the remaining 8 sgRNAs by using RFLP (Appendix A) and used the guide RNA that showed the highest efficiency to generate stable lines. HRMA was used for rapid genotyping of genome-edited zebrafish.

### 2.4. Design and Production of TALENs

The TALE-NT tool was used to design TALEN pairs to target the zebrafish *fus* gene [18,19]. The results from TALE-NT were further filtered by availability of unique restriction sites in the spacer region, repeat array length of 15–17 nucleotides and spacer length of 14–16 nucleotides (Appendix A). Two TALENs designed to target exon 9 of the zebrafish fus gene were assembled by utilizing the ‘golden gate’ approach detailed in Cermak et al. 2011. Subsequently, TALEN mRNA was synthesized via an in vitro transcription process employing the T7 mMESSAGE mMACHINE Kit (Ambion). Each TALEN heterodimer, encoded by 100 picograms of mRNA, was introduced into the cytoplasm of one-cell-stage zebrafish embryos of the wild-type strain [19]. The detailed protocol is provided in Appendix A. The primers used in the production of TALENs and genotyping are provided in Appendix A.

### 2.5. Microinjection

On the day of injections, the injection mix was prepared as follows (Table 1 and Table 2):

Zebrafish TAB WT embryos were collected and the components required for injection were thoroughly mixed and allowed to incubate at room temperature for 5 min, resulting in the formation of a complex, which was subsequently stored in ice. The prepared injection mixture was then loaded into the needle and microinjected into zygotes using established zebrafish injection protocols. Two nanoliters of the injection mixture was delivered directly into the single cell, avoiding the yolk. The injected eggs were cultured in 1× egg water, placed within 100 mm plastic Petri dishes and maintained in an incubator set at a temperature of 28 °C. It was ensured that the embryo density did not exceed 60 embryos per 25 mL of egg water per Petri dish. For comparative purposes, uninjected embryos, constituting the control group, were also obtained from the same clutch and raised at a constant temperature of 28 °C. These embryos were collected for genotyping at 48–72 h post-fertilization (hpf).

### 2.6. Genotyping by RFLP

The successful hits from HRMA assay were further genotyped by restriction-fragment-length polymorphism (RFLP) assay. In this assay, a site-specific restriction enzyme that recognizes the double-strand break was used to detect indels. The failure of restriction enzyme digestion suggests the occurrence of DNA sequence mutations in the target region. The positive hits from RFLP assay were further analysed by cloning, before Sanger sequencing was used to confirm the mutations. RFLP was also utilised as a rapid screening method when the specific change in the target region (whether it created a new restriction site or deleted an existing restriction site) was known. 

### 2.7. Genotyping by Sequencing

Sanger sequencing was used to determine the mutations introduced to the target sequence. Both PCR fragments and plasmids were sequenced by Sanger sequencing. Apart from generic sequencing primers, such as M13 forward, M13 reverse, T7 and SP6, specific sequencing primers flanking the target region were designed using the Primer3Plus tool (http://primer3plus.com/cgi-bin/dev/primer3plus.cgi (accessed on 1 May 2017)). 

Samples were sequenced by Garvan Molecular Genetics Facility and Macrogen Korea using ABI3130x and ABI3730XL instruments, respectively. Sequence trace files were analysed using SnapGene software 1.5.

### 2.8. Immunohistochemistry

Embryos and dissected brains were transferred to microcentrifuge tubes containing 4% PFA and fixed overnight at 4 °C. Following this fixation period, the PFA solution was carefully removed, and the specimens were rinsed in PBST, followed by three additional 5-min washes at room temperature. For whole-mount immunofluorescence staining, these specimens were subjected to a sequence of methanol washes, progressively transitioning from 25% methanol in PBST to 50%, 75%, and ultimately 100% methanol. Following these methanol washes, the samples were carefully stored in 100% methanol at a temperature of −20 °C until they were ready for further use.

### 2.9. Motor Neuron Analysis

Whole mount larval sections were immunostained with SV2 (synaptic vesicle glycoprotein) (1:200). After whole-mount immunofluorescence staining with SV2 antibodies (specifically staining caudal primary motor neurons), spinal motor neuron acorns were analysed.

### 2.10. Touch-Evoked Escape Response

Two zebrafish embryos (48 hpf) per well were plated in 48-well glass plates for behavioral assays (400 μL per well). This density enables manual manipulation of zebrafish larva with a probe. H_2_O_2_ or DTT were added to a final concentration 0.1–312,500 nM, and covered with oxygen permeable seals. Following incubation for 24 h at 28.5 °C, each zebrafish was lightly probed at the top of the tail to stimulate a touch-evoked escape response. The extent of escape response was scored as described in Zhang et al. [20]. 

### 2.11. Locomotion Analysis

To conduct locomotion analysis, we employed the ZebraLab tracking system, comprising a ZebraBox for monitoring zebrafish embryos and larvae and a ZebraTower for tracking adult fish (Viewpoint). In order to specifically track 6-day post-fertilization (dpf) larvae, we placed them in a 24-multi-well plate within a ZebraBox equipped with enhanced lighting conditions. The tracking procedure involved a cycle of 6 min of light, followed by 4 min of darkness, and then another 4 min of light. During this time, the locomotion of the larvae was recorded, with the total distances traveled by each larva within the dark phase calculated (with the first light phase serving as an acclimatization period). The locomotion analysis was conducted at the 6 dpf stage.

To further investigate swimming responses, larvae were transferred to 24-well plates with one larva per well, and placed in 1 ml of E3 medium. Utilizing a high-sensitivity digital camera (30 frames per second) in conjunction with the ZebraLab tracking software (https://www.viewpoint.fr/product/zebrafish/fish-behavior-monitoring/zebralab, accessed on 27 December 2023), we assessed if, there were changes in swimming behavior in response to a stimulus (darkness) in mutant larvae compared to their wildtype counterparts. Movement was monitored during alternating cycles of 15 min of 100% darkness, followed by 15 min of 100% light.

### 2.12. Statistical Methods

In this study, quantitative data were derived from at least three independent experiments or individuals. Descriptive statistics are provided as the mean ± standard error of difference (SE of difference) for data pertaining to ‘n’ individuals or ‘n’ independent experiments. The statistical analyses, including unpaired two-tailed Student’s *t*-tests and one-way ANOVA, were carried out using GraphPad Prism 6.0 software (GraphPad Software) to assess the significance of observed differences. Asterisks refer to the following significance thresholds: *p* < 0.05 (*), *p* < 0.01 (**) and *p* < 0.001 (***).

## 3. Results

### 3.1. Zebrafish Model of Ccnf Deficiency

To determine the suitability of zebrafish as a model to study CCNF, we aligned the human and zebrafish Cyclin F genes (Figure 1A). While zebrafish Ccnf has 61% nucleotide and 53% amino acid identity with respect to human CCNF (Figure 1B), the functional domains are well conserved. In particular, pairwise comparison of human and zebrafish amino acid sequences shows a significant similarity starting at the early PEST sequence (Figure 1C), which harbours two recently identified ALS-associated mutations (S621G, E521X; [1]. We identified the phosphorylation sites in human *CCNF* using NetPhos 3.1, a neural network predictor of phosphorylation sites [21,22]. Of the 36 predicted phosphorylation sites, more than half of these sites (28 serine, 8 threonine sites) exclusively cluster within the PEST domain (Appendix A). Interestingly, 8 of these sites were around the newly identified S621G mutation (1). Therefore, to further investigate the role of CCNF in ALS pathology, we generated a premature stop codon in the PEST domain. 

To generate a *ccnf*-deficient zebrafish model we targeted the start of the PEST domain using CRISPR/Cas9 (Figure 2A,B). We raised injected embryos to adulthood (P0) and screened their progeny (F1) for the desired genomic modifications using HRMA (Figure 2C) and RFLP analysis of PCR amplicons (Figure 2D). A total of 11 of the 84 embryos raised to adulthood successfully transmitted indels in the PEST domain to the F1 (13% efficiency). Of the 11 F1 progeny showing indels (Appendix A), we focused on one line with a frameshift creating a stop codon at the target site. Sequencing confirmed the introduction of a 14bp-long insertion at position 623, resulting in a premature stop codon (Figure 2B). We used HRMA, RFLP and sequencing to classify the wild-type, heteroyzygous and homozygous *ccnf* mutants within the F2 progeny (Figure 2C,D). We observed the nonsense-mediated decay of mutant mRNA transcript reducing mRNA expression in both heterozygous and homozygous *ccnf* mutants (hereafter referred to as *ccnf*
^+/−^ and *ccnf*
^−/−^, respectively) (Figure 2E).

### 3.2. Zebrafish Model of FUS Deficiency

To identify which FUS functions are shared with other ALS-associated genes, we generated a zebrafish model of FUS deficiency. Human and zebrafish Fus proteins share approximately 60% amino acid sequence identity with evolutionary conserved domains (Figure 3A,B). The C-terminal domains, including the RNA recognition motif (RRM), second and third RGG-rich regions, zinc finger domain and nuclear localisation signal (NLS), are particularly well conserved (80–90%) (Figure 3C). Strikingly, the majority of the *FUS* variants that are found in familial ALS patients cluster in the C-terminal domain, particularly within the NLS (Figure 3D). Therefore, to investigate the role of FUS in ALS neuropathology, we targeted the highly conserved C-terminus of zebrafish FUS (Figure 4A,B).

We designed and injected TALEN RNAs, targeting the early RRM domain of the zebrafish *fus* (Figure 4A,B). Of the 38 embryos raised to adulthood, 9 successfully transmitted indels to the F1 progeny, showing 24% germline transmission (Appendix A). We identified carriers with mutations in the targeted region using HRMA, which were then confirmed by Sanger sequencing. Of the F1 progeny showing desired indels, we identified a line with 10 bp deletion after amino acid position 317, resulting in a frameshift mutation and a premature stop codon at position 324 (Figure 4B). We used HRMA, RFLP (Appendix A) and sequencing to identify the wild-type, heteroyzygous and homozygous FUS mutants within the F2 progeny (Figure 4C,D). As a result of nonsense-mediated decay of mutant mRNA transcript, both heterozygous and homozygous FUS mutants (hereafter referred to as *fus*^+/−^ and *fus*^−/−^, respectively) showed reduced levels of *fus* mRNA (Figure 4E).

### 3.3. Phenotypic Characterization of ccnf- and fus-Deficient Zebrafish 

Both *ccnf* and *fus* mutant zebrafish embryos had a morphologically normal appearance and no significant differences in embryo lengths (Appendix A). Previous overexpression and morpholino studies showed that zebrafish *ccnf* [24], and *fus* morphants [25] display significant defects in primary motor neuron morphology. These defects were characterised by severe aberrant branching and/or shortening of axons of motor neurons. To assess if these defects are observed in heterozygous and homozygous *ccnf* knockout embryos, we stained the 2dpf zebrafish embryos with SV2 antibody. Qualitative assessment of motor neurons did not reveal any major differences between homozygous *ccnf* mutants (*ccnf*
^−/−^), heterozygotes (*ccnf*
^+/−^) and wild-type controls. However, blinded quantitative assessment of the motor neuron axon showed a modest but statistically significant decrease in the length of primary motor neuron axons in homozygotes, compared to wild-type controls (Figure 5A; *ccnf*
^+/+^, 199.82 ± 17.43; *ccnf*
^+/−^, 183.15 ± 21.23, *p* = 0.0008).

Similarly, *fus* mutant motor neurons did not display qualitative defects when compared to wild-type controls. In contrast to *ccnf* mutants, homozygous *fus* motor neurons did not display any significant quantitative length defects (Figure 5B; *fus*^+/+^, 189.10 ± 17.78; *fus*^+/−^, 186.80 ± 15.30, *fus*^−/−^, 178.43 ± 22.60 *p* = 0.0811).

### 3.4. Stress Specific Defects in ccnf-Deficient Zebrafish

Oxidative or endoplasmic reticulum (ER) stress elicits a pathological stress response in *fus* mutant cells. Previous studies implied oxidative stress and ER stress potentially had a role in ALS pathology [26,27,28,29,30,31,32].

To assess motor activity of zebrafish embryos, we performed touch-evoked escape response (TEER) assays, in which healthy zebrafish embryos show an escape (swimming) response when touched on the tail. Embryos with impaired motor activity display reduced or no response. Under basal conditions, both *ccnf* and *fus* mutants did not show any obvious defect in motor activity, compared to the wild type controls. To determine if *ccnf* or *fus* mutants were sensitised to oxidative stress, we established a TEER assay with H_2_O_2_ (Figure 6). While *ccnf* deficiency was not exacerbated by 24 h treatment with H_2_O_2_ (Figure 6A), *fus* mutants displayed reduced TEER after treatment with H_2_O_2_, as expected (Figure 6C) [33].

In contrast, both mutants were similarly sensitised to ER stress induced by 24 h treatment with DTT, which resulted in reduced TEER at doses that did not impair *ccnf*
^+/−^ and *fus*^+/−^ mutants or wild-type controls. (Figure 6B,D). Although FUS is required for the response to both stressors, CCNF appears to be specifically required for the ER stress response.

### 3.5. ccnf-Deficient Zebrafish Embryos Display Reduced Photomotor Response

To further assess more complex motor functions in zebrafish embryos, we used the photo-motor response (PMR) screening assay. This method measures a range of motor behaviours by subjecting 6 dpf embryos, placed in 24-well plates, to a 4-min darkness response test alternating between light and dark cycles (Appendix A). Only offspring from the same clutch were used in each test to control for potential genetic and/or batch effects (*n* = 287 and *n* = 440 for ccnf and fus, respectively). 

Both ccnf and fus mutant zebrafish displayed diminished activity patterns during the PMR assay. In the ccnf mutants, the swimming trajectories showed reduced activity, notably in *ccnf*
^−/−^ homozygotes (Appendix A), which were significantly lower than wild-type. (Figure 7A,B; *ccnf*
^−/−^, distance 867.27 mm ± 26.51; velocity: 3.61 mm/s ± 0.11, *n* = 60; *ccnf*
^+/+^, distance 1213.61 mm ± 18.57; velocity: 5.06 mm/s ± 0.08, *n* = 89, *p* < 0.0001). This aligns with our observations of the TEER responses to ER-stress, suggesting the lack of the PEST region substantially dampens the photomotor response in zebrafish embryos. Interestingly, *ccnf*
^+/−^ mutants also displayed a reduced motor activity, compared to the wild-type controls (Figure 7A,B; *ccnf*
^+/−^, distance 1046.19 mm ± 15.02; velocity: 4.36 mm/s ± 0.06, *n* = 138; *ccnf*
^+/+^, distance 1213.61 mm ± 18.57; velocity: 5.06 mm/s ± 0.08, *n* = 89, *p* < 0.0001). Post-hoc analysis also revealed that *ccnf*
^−/−^ mutants travelled shorter distances than heterozygous *ccnf*
^+/−^ mutants, indicating the lack of PEST region significantly impairs the photomotor response in zebrafish embryos (*p* < 0.01).

Similarly, the fus mutants, and particularly the *fus*
^−/−^ group, exhibited reduced activity in their swimming trajectories (Figure 7C,D; *fus*^−/−^, distance 874.47 mm ± 28.39; velocity: 3.64 mm/s ± 0.12, *n* = 107; *fus*^+/+^, distance 1164.57 mm ± 21.92; velocity: 4.85 mm/s ± 0.09, *n* = 112, *p* < 0.001 Appendix A). However, unlike the *ccnf* mutants, the *fus^+/**−**^* group did not exhibit any significant reduction in motor activity when compared to the wild-type. (Figure 7C,D; *fus*^+/−^, distance 1143.67 mm ± 14.68; velocity: 4.77 mm/s ± 0.06, *n* = 221; *fus*^+/+^, distance 1164.57 mm ± 21.92; velocity: 4.85 mm/s ± 0.09, *n* = 112, *p* = 0.7369).

## 4. Discussion

We employed genome editing tools, namely the CRISPR/Cas9 system for CCNF and the TALEN system for FUS, to study the phenotypic consequences of gene disruptions on ALS-relevant whole organism phenotypes.

Genome editing tools such as CRISPR/Cas9 and TALENs have revolutionized the field of molecular biology by enabling precise modifications of the genome. These tools offer unprecedented opportunities to investigate gene function in vivo, allowing researchers to study the consequences of gene disruptions, deletions, or modifications. In the context of neurodegenerative diseases like ALS, genome editing tools provide a powerful means to create animal models that closely resemble human pathology and uncover the functional impact of disease-associated gene mutations.

Although the use of CRISPR/Cas9 has become ubiquitous in genome editing, TALENs remain a powerful system for generating disease models. CRISPR/Cas9 and TALEN systems show both advantages and limitations when applied in genome editing studies. The CRISPR/Cas9 system offers simplicity and is versatility, and relatively low cost, and the TALEN system provides a more customizable approach with potentially higher specificity [34]. The applications of different genome editing strategies are discussed in greater detail by Denes et al. [35].

### 4.1. Phenotypic Characterization of CCNF and FUS Mutants

While ccnf-deficient zebrafish displayed no significant morphological defects during embryonic development, further analyses revealed alterations in motor behaviors. Furthermore, in the context of the stress response, ccnf-deficient zebrafish exhibited reduced motor activity, specifically when endoplasmic reticulum (ER) stress was induced by DTT treatment. Interestingly, oxidative stress induced by H_2_O_2_ did not significantly affect the motor activity of ccnf mutants. These observations suggest that CCNF may play a specific role in protecting against ER stress-induced motor impairments. The differential response to stressors highlights the complexity of stress pathways and the need for further investigations with a larger panel of diverse stressors, as this will elucidate the precise mechanisms through which CCNF influences motor neuron function.

Similarly, *fus*-deficient zebrafish displayed no morphological defects during embryonic development, which is consistent with previous studies that used genome editing to disrupt the *fus* gene in zebrafish [33]. These genetic models allow the anaylsis of naturally bred and raised embryos. Therefore, the stronger developmental abnormalities observed in morphants may be due to the mechanical stress of microinjection or the off target/nonspecific effects of morpholino reagents [24,33,36]. The lack of morphological abnormalities indicates that FUS may not be essential for early zebrafish development. However, phenotypic analyses revealed motor impairments in *fus* mutants and photomotor response assays demonstrated compromised locomotion activity. Both oxidative stress (H_2_O_2_) and ER stress (DTT) treatments disrupted motor activity in *fus* mutants. These findings suggest that FUS is involved in stress response pathways and plays a role in maintaining proper motor neuron function in challenging conditions. In our study, the potential for nonsense-mediated decay is considered, especially in relation to the RNA stability (Figure 2E and Figure 4E). This cellular mechanism, which targets aberrant mRNA transcripts for degradation, could significantly influence the phenotypic outcomes of the *ccnf* and *fus* gene mutations in our zebrafish models, and offer insight into the broader implications of gene editing for gene expression dynamics.

### 4.2. Comparison of CCNF and FUS Models

Although both the *ccnf* and *fus* mutants exhibited reduced motor activity in stress conditions, there were notable differences in their phenotypic characteristics. The *ccnf* mutants specifically showed impaired motor activity in response to ER stress induced by DTT treatment, while *fus* mutants displayed motor impairments in both oxidative stress (H_2_O_2_) and ER stress conditions. These distinct stress response profiles indicate overlapping yet distinct roles for CCNF and FUS in stress pathways and motor neuron function. Furthermore, the phenotypic changes observed in the zebrafish models were less pronounced when compared to previous studies employing overexpression of mutant proteins [36,37,38]. This highlights the importance of studying gene disruptions in an endogenous context, with the aim of better understanding the true contribution of these genes to disease manifestation. The use of genome editing tools, such as CRISPR/Cas9 and TALENs, allows for more accurate and physiologically relevant investigations of gene function.

### 4.3. Role of Cellular Stress in ALS Pathogenesis

The role of cellular stressors, such as oxidative stress and ER stress, has been implicated in ALS manifestation. Previous studies have reported increased oxidative stress and ER stress in ALS models and patients, suggesting their involvement in motor neuron degeneration. In our study, we utilized oxidative stress and ER stress to elicit motor impairments in zebrafish larva by using the TEER assay. The disrupted TEER activity, in response to stressors in both *ccnf* and *fus* mutants, indicates the sensitivity of motor neurons to cellular stress and highlights the potential role of stress pathways in ALS pathology.

Moreover, our findings support the two-hit hypothesis [39,40], which suggests that a single hit to a disease-associated gene may not be sufficient to manifest significant ALS pathology, meaning that additional stressors or genetic hits may be required to elicit a motor phenotype. The observed motor impairments in the zebrafish mutants in stress conditions demonstrate the importance of stress response pathways in motor neuron function and the potential contribution of stressors to ALS pathology. The stable *fus* deficient models generated in this study may help resolve the complex roles of FUS in ALS, as recognized in a recent review by Aksoy et al. [5]. This is particularly important as most previous studies utilized transient knockdown or overexpression of FUS. 

## 5. Conclusions

In conclusion, our study, which used genome editing tools, demonstrated the functional roles of CCNF and FUS in motor neuron biology, and in particular their involvement in stress response pathways. The zebrafish models featuring disrupted *ccnf* and *fus* genes will be invaluable resources for further explorations of ALS pathology, and have the clear potential to contribute to the development of novel therapeutic interventions. By leveraging the precision of CRISPR/Cas9 and TALEN systems, we have significantly advanced our comprehension of gene function within the context of neurodegenerative diseases.

Our findings underscore the critical role of stressors in ALS pathogenesis. Stress-induced motor impairments observed in both *ccnf* and *fus* mutants underscore the significance of stress response pathways in motor neuron function, and also underline the potential contribution of stressors to ALS-related motor neuron degeneration.

The zebrafish models we have generated serve as promising platforms for identifying potential therapeutic targets and conducting compound screenings that aim to ameliorate the motor phenotype associated with ALS. As we continue to delve into this field, our ongoing research promises to make substantial contributions to the development of effective treatments for ALS and other related disorders, offering hope to those affected by these devastating neurodegenerative diseases.

## Figures and Tables

**Figure 1 cells-13-00372-f001:**
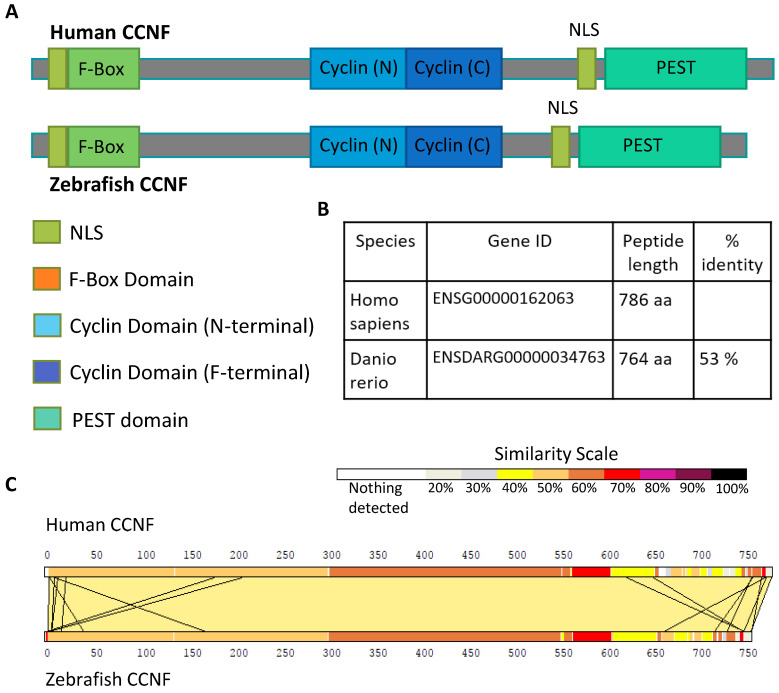
Schematic overview of CCNF protein. (**A**) Graphical alignment of the functional domains in human and zebrafish CCNF proteins. (**B**) Conservation of full-length human and zebrafish CCNF proteins. (**C**) A pair-wise comparison of the sequence similarity of human CCNF and zebrafish Ccnf, displayed by LalnView [23].

**Figure 2 cells-13-00372-f002:**
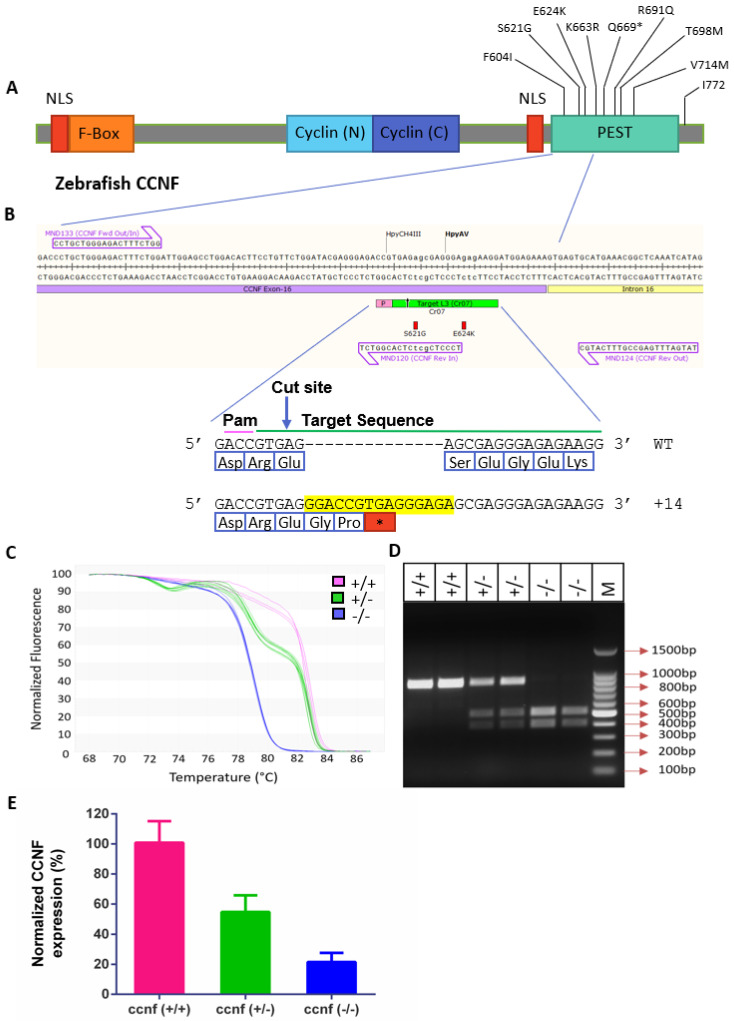
Generating *ccnf*-deficient zebrafish lines. (**A**) Design of sgRNA targeting the PEST region of *ccnf*. (**B**) +14bp insertion at the CRISPR/Cas9 target site resulting with premature stop codon (**C**) High resolution melting analysis (HRMA) curves showing WT (*ccnf* ^+/+^), heterozygous (*ccnf*
^+/−^) and homozygous (*ccnf*
^−/−^) zebrafish lines (**D**) RFLP assay showing the PCR amplicons of *ccnf*
^+/+^*, ccnf*
^+/−^
*and ccnf*
^−/−^ digested with AvaII enzyme. (**E**) Quantification of *ccnf*
^+/−^ and *ccnf*
^−/−^ mRNA, relative to *ccnf* ^+/+^ mRNA in 4dpf embryos.

**Figure 3 cells-13-00372-f003:**
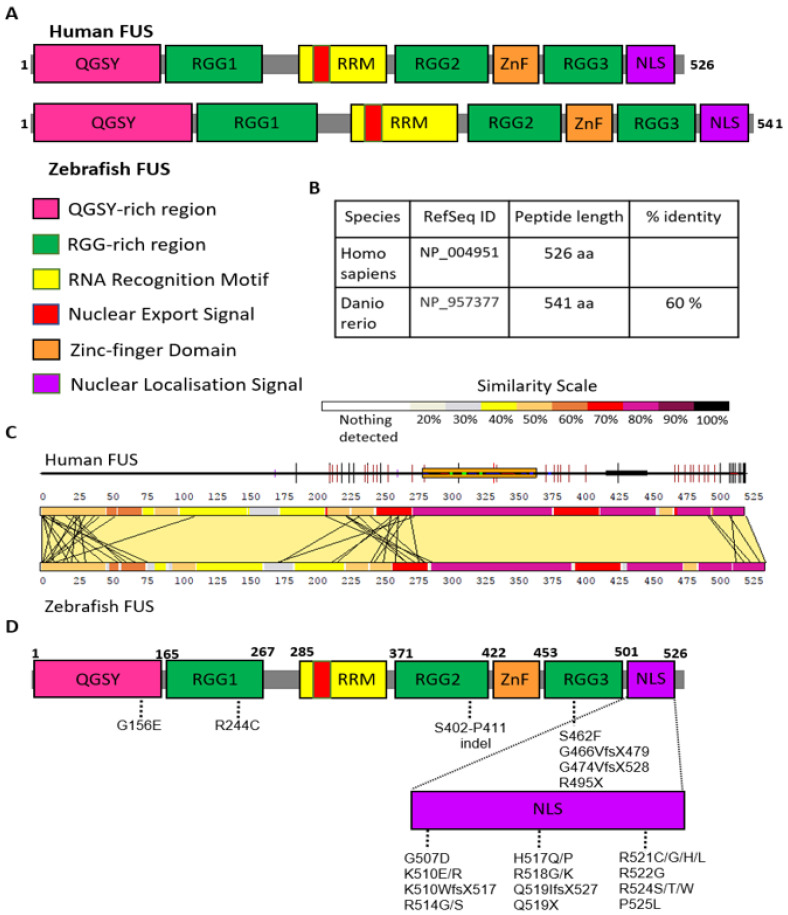
Schematic overview of FUS protein. (**A**) Graphic illustration of the functional domains of human and zebrafish FUS proteins and their locations. (**B**) Homology comparison between two species shows 60% identity in amino acids sequences. (**C**) A pair-wise comparison of the sequence of human FUS and zebrafish Fus displayed by LalnView [8]. (**D**) Location of ALS-associated *FUS* mutations.

**Figure 4 cells-13-00372-f004:**
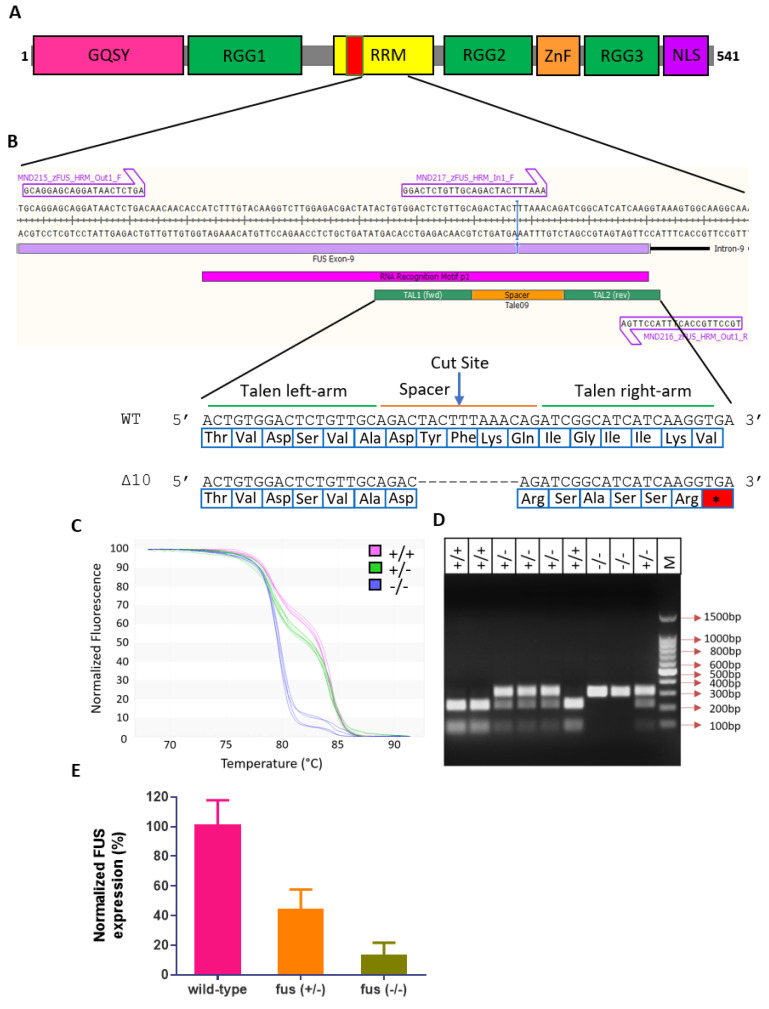
Generating *fus*-deficient zebrafish lines. (**A**) Design of sgRNA targeting the RRM region of *fus*. (**B**) −10bp deletion at the TALEN target site resulting in a premature stop codon (*) (**C**) High resolution melting analysis (HRMA) curves showing WT (*fus^+/+^*), heterozygous (*fus^+/−^*) and homozygous (*fus^−/−^*) zebrafish lines (**D**) RFLP assay showing the PCR amplicons of *fus^+/+^*, *fus^+/−^* and *fus^−/−^* digested with DraI enzyme. (**E**) Quantification of *fus^+/−^* and *fus^−/−^* mRNA, relative to *fus^+/+^* mRNA in 4dpf embryos.

**Figure 5 cells-13-00372-f005:**
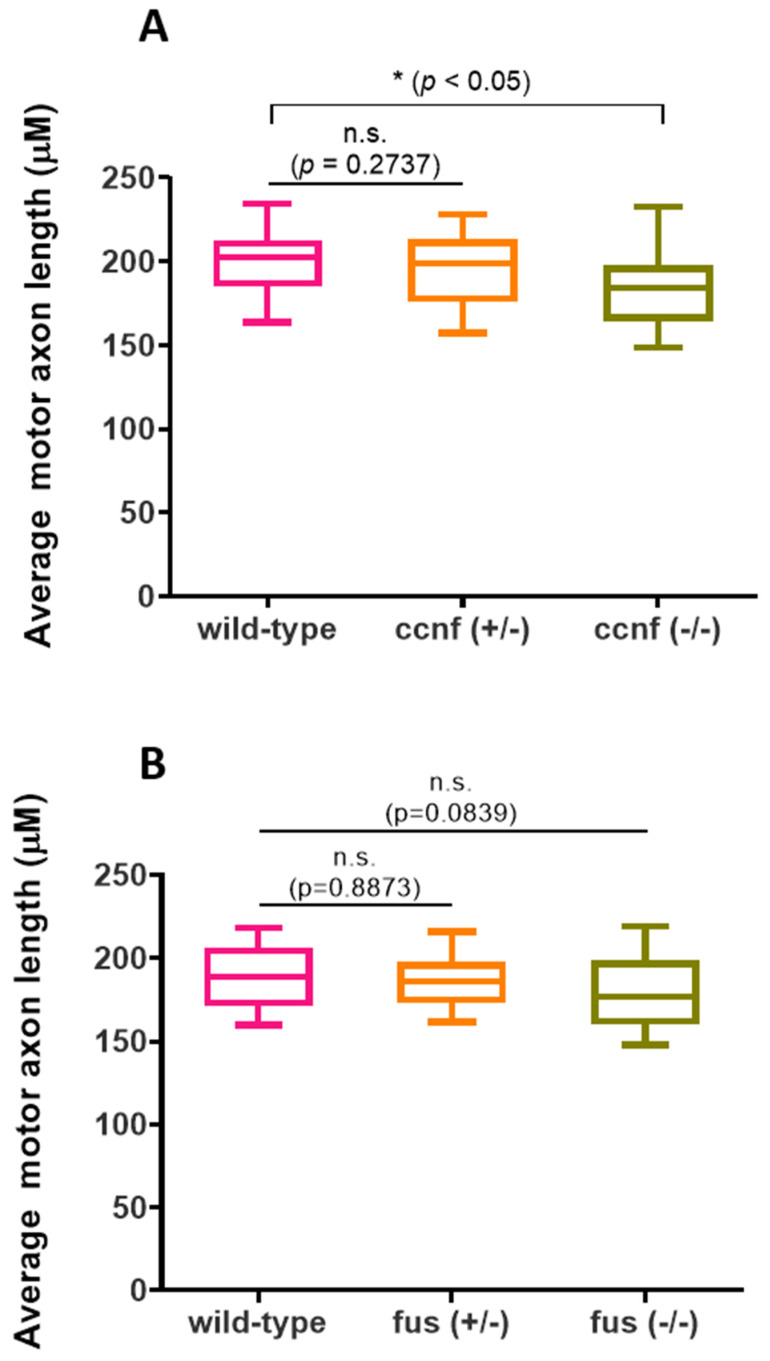
Motor neuron axon growth and length. The average length of six motor neurons above the yolk sac extension were analysed in all three genotypes of (**A**) ccnf (*n* = 41 fish) and (**B**) fus (*n* = 90). Average motor axon length was analysed per genotype derived from three different clutches. Mean + SEM. Statistical test used: Kruskal-Wallis test, followed by Dunn’s multiple comparisons test.

**Figure 6 cells-13-00372-f006:**
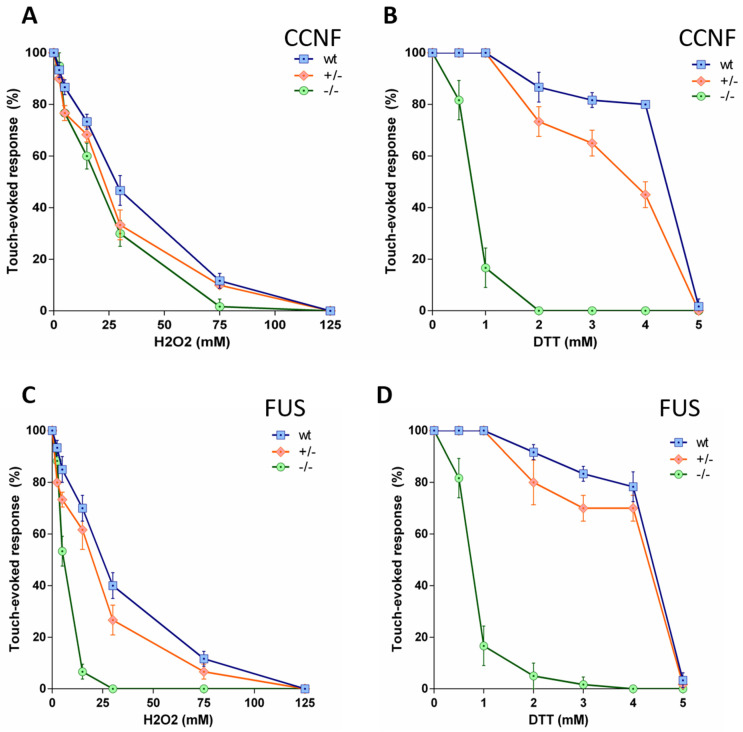
Touch-evoked response of 2dpf zebrafish larvae treated with H_2_O_2_ and DTT. (**A**,**B**) Dose-response curves for the effect of H_2_O_2_ and DTT on the touch-evoked escape response (CCNF; *n* = 24 zebrafish larvae per genotype and dose). (**C**,**D**) Dose-response curves for the effect of H_2_O_2_ and DTT on the touch-evoked escape response (FUS; *n* = 24 zebrafish larvae per genotype and dose).

**Figure 7 cells-13-00372-f007:**
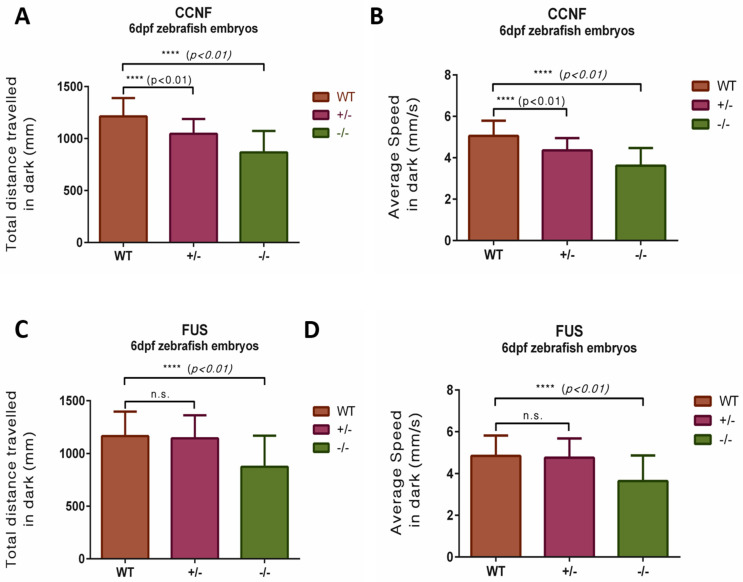
Photomotor response of *ccnf* and *fus* knockout larvae. (**A**,**B**) Quantification of average swimming speed (during 4-min in darkness and total distance swum during the 4-min in darkness) revealed that *ccnf*
^−/−^ travelled shorter distances than both *ccnf*
^+/−^ and *ccnf*
^+/+^. (**C**,**D**) Quantification of average swimming speed (during 4-min in darkness and total distance swum during the 4-min in darkness) revealed that *fus*^−/−^ travelled shorter distances than both *fus*^+/−^ and *fus*^+/+^.

**Table 1 cells-13-00372-t001:** CRISPR/Cas9 injection mix for zebrafish embryos.

	Knock-Out Mix		Knock-In Mix	
Contents	1× (µL)	[Final] (ng/µL)	1× (µL)	[Final] (ng/µL)
Crispr sgRNA	3.5	75	3.5	75
Cas9 Protein	1.0	500	1.0	500
DNA Template	-	-	1	25
Phenol Red	0.5	n/a	0.5	n/a

**Table 2 cells-13-00372-t002:** TALEN injection mix for zebrafish embryos.

	Knock-Out Mix	
Contents	1× (µL)	[Final] (ng/µL)
TALEN Left Arm	2.5	500
TALEN Right Arm	2.5	500
Phenol Red	1.0	n/a

## Data Availability

The raw data supporting the conclusions of this article will be made available by the authors on request.

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
