# Peer review of "Zebrafish CCNF and FUS Mediate Stress-Specific Motor Responses"

_cells, 2024, doi:10.3390/cells13050372_

Round 1

Reviewer 1 Report

Comments and Suggestions for Authors

The authors utilized gene editing techniques, including CRISPR and Zinc Finger nulcase, to perform targeted mutations on two genes in zebrafish. These two genes are Cyclin F and FUS, respectively. The authors optimized the technology and successfully obtained mutants. Furthermore, they conducted phenotype analysis on the mutated fish.

They focus on the axonal growth of motor neurons and the response of larvae to hydrogen peroxide and DTT.

Preliminary analysis of the mutants was conducted, and the entire work was quite good.

The area that needs improvement is the use of immunohistochemistry mentioned in the article and obtaining images, but the main figures do not show the staining or fluorescence images. These data are very important and require representative images.

The analysis of phenotype is slightly weak, and more data support is needed in this regard.

Reviewer 2 Report

Comments and Suggestions for Authors

This manuscript presents the generation of loss of function mutants for the zebrafish ccnf and fus genes. A detailed molecular description and characterization of the mutants is provided as well as a few phenotypic traits mainly having to do with motor axons, touch-evoked response and photomotor response.

 Loss of function studies for these two genes have already been published and are adequately referenced by the authors of the current manuscript. This being said, the mutants produced for the current manuscript constitute a useful resource. Although I am not requesting that the authors redo their CRISPR Cas9 experiment, they should be reminded that a design that leads to the gene deletion is often preferable as it reduces the risk of transcriptional compensation.

 It would also have been interesting to see what happens when the two mutants are bred to each other, although I am not asking the authors to do it.

 Specific comments:

-          Some methodological information is missing, including the RNA expression analysis for the two mutants (Figure 2E, Figure 4 E). Some discussion on nonsense mediated decay should also be added to the text, especially that we are not told about the genomic structure of the two genes.

-          The authors don’t discuss much the use of CRISPR for one gene and TALEN for the other. If there is a good reason for the two difference approaches, it should be mentioned in the discussion. Furthermore, the statement that TALENs proved .a more customizable approach with potentially higher specificity, should be supported/expanded or at least referenced.

Minor comments: 

-          Line 223: a reference should be given (or reiterated) for the S621G mutation.

-          Line 290: Replace the word “mutants” with “morphants”.

-          Line 323: “as expected”. This puzzled me at first but this expectation comes from a similar observation made in the Lebedeva 2017 paper. The reference should be reiterated here.

Reviewer 3 Report

Comments and Suggestions for Authors

This is good paper, well designed. I only recommend to use full names in title. Also please provide more references in the introduction and discussion. You write some sentences and o not provide references (almost all of them, especially in introduction).
